# Introducing a blocked procedure in nonparametric CD-CAT

**Jiahui Zhang[1], Yuqing Yuan[2], Ziying Qiu[2]\*, Feng Li[3]\***

**1** Collaborative Innovation Center of Assessment for Basic Education Quality, Beijing Normal University, Beijing, China, **2** School of Statistics, Beijing Normal University, Beijing, China, **3** Collaborative Innovation Center of Assessment for Basic Education Quality, Beijing Normal University at Zhuhai, Zhuhai, Guangdong, China

\* lifeng@bnu.edu.cn (FL); 598023531@qq.com (ZQ)

## Abstract

Cognitive Diagnostic Computerized Adaptive Testing (CD-CAT), in conjunction with non-parametric methodologies, is an adaptive assessment tool utilized for diagnosing students' knowledge mastery within smaller educational contexts. Expanding upon this framework, this study introduces the blocked procedure previously used in the parametric CD-CAT, enhancing the flexibility of nonparametric CD-CAT by enabling within-block item review and answer modification. A simulation study was conducted to evaluate the performance of this blocked procedure within the context of nonparametric CD-CAT across varied conditions. With increasing block size, there was a marginal reduction in pattern correct classification rate; however, such differences diminished as item quality or test length augmented. Over-all, under a majority of conditions, the blocked procedure, characterized by block sizes of 2 or 4 items, allows item review within-block while attaining satisfactory levels of classification accuracy. The integration of within-block item review and answer modification with nonpara-metric CD-CAT fosters a more adaptive and learner-centric testing environment.

## Introduction

In recent years, Cognitive Diagnosis Assessment (CDA) has garnered widespread interest due to its potential to provide diagnostic feedback by integrating modern measurement models with cognitive psychology frameworks [1]. The combination of Cognitive Diagnosis Models (CDMs) with Computerized Adaptive Testing (CAT) has further adapted to students' individual characteristics, providing diagnostic information through Cognitive Diagnostic Computerized Adaptive Testing (CD-CAT), thus enhancing the efficiency of assessment and feedback mechanisms [2]. Moreover, it has been observed to foster increased student engagement and motivation [3].

Formative assessment and immediate feedback are vital for enhancing learning [4–6]. Classroom assessment, when integrated with measurement technologies such as CDMs and CAT, can better serve a diagnostic function, providing prompt feedback to support both student learning and teacher instruction. To address the specific needs of small-scale educational settings like classrooms, several studies have explored nonparametric approaches to CDA [7–

**Data Availability Statement:** The data and code files for the simulation study are available in the Open Science Framework repository (https://osf.io/m4fce/).

**Funding:** This research was supported by the National Key R&D Program of China

(2021YFC3340801). The funders had no role in study design, data collection and analysis, decision to publish, or preparation of the manuscript.

**Competing interests:** The authors have declared that no competing interests exist.

9], which do not rely on parametric statistical models. Some researchers have applied nonparametric approaches to CD-CAT to improve the efficiency of classroom formative assessment. For example, Chang et al. introduced a nonparametric item selection (NPS) strategy, using the nonparametric classification method (NPC) to estimate student mastery patterns [8, 9]. Following this line of research, Chiu and Chang developed a more flexible method, the generalized nonparametric item selection strategy (GNPS), utilizing the generalized nonparametric classification method (GNPC) for estimating student mastery patterns [10, 11].

Another line of research addresses the limitation of CD-CAT relating to examinees' inability to review items and modify their answers. For example, Kaplan and de la Torre proposed a blocked procedure for parametric CD-CAT. This innovation enables examinees to review and modify answers within each item block. Leveraging an on-the-fly assembled multistage adaptive testing (OMST) method, this approach dynamically assembles the next set of items based on examinees' prior responses [12]. Moreover, it serves to mitigate the impact of measurement errors stemming from test anxiety and other factors [13].

This study integrates the benefits of both nonparametric item selection strategies and the blocked procedure, proposing a nonparametric blocked CD-CAT design with answer modification capability. The proposed design introduces the blocked procedure to small-scale classroom settings, enhancing its practicality as a tool for formative assessment. This approach enables teachers to effectively identify individual student issues in daily assignments and classroom quizzes, and thereby maximizes the practical utility of cognitive diagnostic models.

The subsequent sections of this paper are organized as follows. First, a review of CAT designs that allow item review is provided, with a focus on the blocked CD-CAT procedure. Following that, the specific research questions and key techniques of this study are introduced, detailing on how nonparametric item selection strategies are utilized to achieve online assembly within blocks. Subsequently, simulation studies aimed at illustrating performance under various testing conditions are presented. Finally, the results of the simulation studies are discussed, along with limitations of the research and suggestions for future directions.

## CAT with item review

Traditional CAT operates at the item level: after each response, the examinee's estimated ability level (in IRT) or mastery pattern (in CDM) is updated, based on which the next item is selected [11, 14]. Examinees must complete all items sequentially and are unable to review or modify their responses throughout the test, which contradicts examinees' accustomed test-taking behaviors and thus may increase psychological stress and test anxiety [15]. Permitting examinees the opportunity to review and modify their answers during the test could alleviate test anxiety to some extent, enabling correction of errors stemming from carelessness (e.g., spelling errors), misinterpretation of questions, momentary memory lapses, or concentration deficits, thus improving test validity [15, 16]. Consequently, researchers have proposed CAT designs allowing item review. From the perspective of CAT procedures, there are two approaches to implementing item review: one is based on traditional item-level CAT, maintaining testing at the item level while restricting opportunities for answer review and modification [15, 17]; the other follows the concept of Multistage Testing (MST), conducting testing in multiple stages, also referred to as blocked CAT [13, 18].

Taking the item-level approach, Stocking proposed three designs for item review in IRT--CAT [15]. The first design allows examinees to modify answers to a limited number of items after completing the test. The second design divides the item-level CAT into multiple sections, permitting examinees to review and modify answers within each section; once they proceed to the next section, they cannot return to the previous sections. This method is also referred to as

the Successive Block Method. The third design allows examinees to review items within a testlet, which is a group of items associated with the same "stimulus". Previous research has shown that the first two methods effectively reduce the influence of examinees' use of Wainer's strategy by limiting review opportunities, demonstrating good performance in terms of estimated bias and standard errors [15]. Gao et al. introduced Reviewable Cognitive Diagnostic Computerized Adaptive Testing (RCD-CAT), which allows item review, extending Stocking's first two designs to the context of CDA [19]. They found that both designs improved classification accuracy compared to traditional item-level CD-CAT, with the Successive Block Method showing superior effectiveness. The item review designs based on Stocking's proposals add complexity to the original item-level CAT. More importantly, examinees must pay extra attention to the rules on item review, such as the number of attempts allowed and the timing for starting reviews.

The MST represents an alternative approach to item review. In comparison to traditional CAT, the MST offers advantages such as item review [20], flexibility in the number and length of test stages, easiness to satisfy non-statistical constraints [21], and faster processing speed [22]. In an MST design, examinees can review items and modify their answers within each stage or block. There is no limit to the number of attempts or the timing of reviews within a stage or block. In order to further enhance MST, Zheng and Chang proposed On-the-Fly Assembled Multistage Adaptive Testing (OMST) in the context of IRT, where the assembly of the next stage's items is based on the results of previous stages [13]. Two studies introduced the concept of on-the-fly MST to the context of CDA [12, 23], which inspired the present study. These studies are described in more detail in the next section.

## The blocked procedure for parametric CD-CAT

Students' mastery of multiple knowledge points, skills, or other latent traits, referred to as attributes, is often of interest in CDA [24, 25]. The combination of mastery states of these attributes is referred to as the attribute mastery pattern (AMP) or knowledge state, often represented as a vector of binary elements, with '1' indicating mastery and '0' indicating non-mastery [26]. The relationship between the items and the attributes is described in an indicator matrix, called the Q matrix, which has rows corresponding to items, columns corresponding to attributes, and binary elements indicating whether an attribute is measured by an item (that is, whether mastery of an attribute is required to succeed on an item) [24]. The row vectors of the Q-matrix are also referred to as q-vectors, which are usually pre-identified for each item and expressed as vectors of '1's and '0's. A q-vector of (0, 1, 0) indicates that only the second attribute is required to respond to the item [24].

As we mentioned earlier, two studies introduced MST to CDA. In the first study, Liu et al. proposed CD-OMST (on-the-fly assembled multistage adaptive testing to cognitive diagnosis) [23]. Their study provided comprehensive details of the item assembly methods within CD-OMST modules, categorizing assembly methods based on the presence of non-statistical constraints in the test. In the other study, Kaplan and de la Torre independently introduced OMST to the cognitive diagnosis field, and proposed the blocked-CAT procedure for CD-CAT [12]. In contrast to Liu et al., who focused on the testing scenarios with complicated non-statistical constraints, the blocked-CAT procedure primarily addresses simple application scenarios, such as classroom quizzes and unit tests [12, 23].

Specifically, Kaplan and de la Torre introduced different versions of the blocked procedure, imposing different constraints on the q-vectors within a block. The unconstrained version selects items solely based on the information criterion, while the constrained version requires that the items within the same block have different q-vectors. Their simulation study found that the constrained version improved classification accuracy rate; using PWKL as the item selection strategy, increasing block size led to substantially lower classification rates; and when

using other item selection strategies (e.g., MPWKL), increasing the size of the blocks led to negligible decrease in classification rates for low-quality items with longer test lengths and for high-quality items regardless of their test lengths [12].

Overall, for low-stakes testing situations for diagnosing learning outcomes, the blocked CD-CAT can overcome the drawbacks of conventional non-modifiable CD-CAT by incorporating the capability for item review and answer modification, while maintaining relatively low algorithmic complexity. However, both studies are based on parameterized CD-CAT, requiring large samples for calibrating item parameters to guarantee test accuracy [12, 23].

## The proposed blocked procedures

The nonparametric CD-CAT methods offer a measurement approach to achieve intelligent diagnosis in small-scale educational settings [8–11, 27]. Nevertheless, a limitation of current nonparametric CD-CAT is the absence of an option for examinees to review items and revise answers, which contradicts examinees' established testing practices. Current designs that permit item review have been set in the context of parametric CD-CAT [12, 19].

Therefore, this paper introduces the blocked procedure to the nonparametric CD-CAT, referred to as the nonparametric blocked CD-CAT. The objective is to leverage the strengths of both the nonparametric method and the blocked procedure. The proposed blocked procedures, referred to as blocked NPS and blocked GNPS, extend the item selection strategies of item-level CD-CAT, NPS and GNPS [9, 11].

### The blocked procedures

The blocked NPS and the blocked GNPS define item discrimination power differently and employ distinct methods to estimate the examinee's attribute mastery pattern (i.e., NPC or GNPC).

The ideal responses, as a central feature of both NPC and GNPC, are described first. The NPC method define two types of ideal responses to each item, corresponding to two extreme conditions of the relationship between the q-vector and the attribute mastery pattern: conjunctive and disjunctive [8]. The conjunctive ideal response to item $j$ for mastery pattern $\boldsymbol{\alpha}$, denoted as $\eta_j^{(c)}(\boldsymbol{\alpha})$, is defined by the deterministic input noisy output "AND" gate (DINA) model [28, 29]. The disjunctive ideal response to item $j$ for mastery pattern $\boldsymbol{\alpha}$, denoted as $\eta_j^{(d)}(\boldsymbol{\alpha})$, is defined by the deterministic input noisy output "OR" gate (DINO) model [30]. Considering the various conditions in between the two extremes of conjunctive and disjunctive assumptions, the GNPC method introduces the weight $w_{uj}$ given mastery pattern $\boldsymbol{\alpha}_u$ and item $j$ and defines the weighted ideal response pattern for mastery pattern $\boldsymbol{\alpha}_u$ as

$$\eta_j^{(w)}(\boldsymbol{\alpha}_u) = w_{uj}\eta_j^{(c)}(\boldsymbol{\alpha}_u) + (1 - w_{uj})\eta_j^{(d)}(\boldsymbol{\alpha}_u), \tag{1}$$

where $0 \leq w_{uj} \leq 1$ [10].

In both NPC and GNPC, we denote the most likely mastery pattern and the second most likely mastery pattern as $\hat{\boldsymbol{\alpha}}_i^{(t)}$ and $\tilde{\boldsymbol{\alpha}}_i^{(t)}$, respectively, after $t$ items haven been administered.

**Blocked NPS.** The blocked NPS, like the original NPS method, uses the NPC method for mastery pattern estimation [8, 9]. If the items conform to a conjunctive mode, the NPS method defines the discrimination power of item $j$ as the absolute difference between $\eta^{(c)}(\tilde{\boldsymbol{\alpha}}_i^{(t-1)})$ and $\eta^{(c)}(\hat{\boldsymbol{\alpha}}_i^{(t-1)})$ when forming the next block for examinee $i$ given $t$ items have been administered:

$$d_j(\eta_j^{(c)}(\tilde{\boldsymbol{\alpha}}_i^{(t)}), \eta_j^{(c)}(\hat{\boldsymbol{\alpha}}_i^{(t)})) = |\eta_j^{(c)}(\tilde{\boldsymbol{\alpha}}_i^{(t)}) - \eta_j^{(c)}(\hat{\boldsymbol{\alpha}}_i^{(t)})|. \tag{2}$$

Eq 1 can also be formulated using the disjunctive ideal response $\eta^{(d)}(\boldsymbol{\alpha}_u)$. The unconstrained blocked NPS identifies items with non-zero $d(\eta^{(c)}(\tilde{\boldsymbol{\alpha}}_i^{(t)}), \eta^{(c)}(\hat{\boldsymbol{\alpha}}_i^{(t)}))$ within the item pool $R^{(t)}$ at time $t$. These items are considered as more discriminating than others. From this subset of items, $J_s$ items are then randomly selected to form the next block. No item calibration is needed in blocked NPS.

**Blocked GNPS.** The blocked GNPS, following the original GNPS method, uses the GNPC method for mastery pattern estimation [10, 11]. When forming the next block for examinee $i$ given $t$ items have been administered, the discrimination power of item $j$ is defined as

$$d_j(\hat{\eta}_j^{(w)}(\tilde{\boldsymbol{\alpha}}_i^{(t-1)}), \hat{\eta}_j^{(w)}(\hat{\boldsymbol{\alpha}}_i^{(t-1)})) = |\hat{\eta}_j^{(w)}(\tilde{\boldsymbol{\alpha}}_i^{(t-1)}) - \hat{\eta}_j^{(w)}(\hat{\boldsymbol{\alpha}}_i^{(t-1)})|. \tag{3}$$

where $\hat{\eta}_j^{(w)}(\boldsymbol{\alpha}_u) = \hat{w}_{uj}\eta_j^{(c)}(\boldsymbol{\alpha}_u) + (1 - \hat{w}_{uj})\eta_j^{(d)}(\boldsymbol{\alpha}_u)$. The weights $\hat{w}_{uj}$ are estimated from the response data of the calibration sample, thus $\hat{w}_{uj}$ adapts automatically to the data's complexity and variability [10]. Therefore, there is no need to know the underlying CDM. The unconstrained blocked GNPS then selects the $J_s$ items with the highest the discrimination power $d(\hat{\eta}^{(w)}(\tilde{\boldsymbol{\alpha}}_i^{(t-1)}), \hat{\eta}^{(w)}(\hat{\boldsymbol{\alpha}}_i^{(t-1)}))$ from the current item pool $R^{(t)}$ for inclusion in the next block.

## Constraining the q-vectors

We distinguish between two versions of the blocked procedure as illustrated in Fig 1: the left panel depicts the unconstrained version of the proposed procedure described in the previous section, and the right panel presents the constrained version, which imposes constraints on the q-vectors within blocks. Both the blocked NPS and blocked GNPS methods can be implemented in either an unconstrained or a constrained version.

The constrained version of the blocked procedure aims to make the q-vectors of items within the same block as distinct as possible. The rationale behind introducing the constrained version lies in its potential improvement on the performance of the adaptive testing procedure. Previous research in parametric methods indicated that administering items with the same q-vector repeatedly did not provide additional information for improving measurement precision [31]. Kaplan and de la Torre introduced the constrained version to their parametric blocked procedure to enhance the diagnostic power of the test by avoiding redundancy and ensuring a broader coverage of the attribute space [12].

**The constrained version of the blocked NPS.** First, determine the number of q-vector types, denoted as $M$, by counting the distinct q-vectors in the subset of items with non-zero $d(\eta^{(c)}(\tilde{\boldsymbol{\alpha}}_i^{(t)}), \eta^{(c)}(\hat{\boldsymbol{\alpha}}_i^{(t)}))$ (see Eq 2) within the item pool $R^{(t)}$ at time $t$. Second, if $\geq J_s$, randomly select $J_s$ q-vector types and then randomly choose one item from each type to form the next block. If $M < J_s$, distribute the required $J_s$ items as evenly as possible across the $M$ q-vector types and randomly select items from each type. While it may not be possible to ensure all q-vectors are unique, they should be as distinct as possible, though some repetition may occur.

**The constrained version of the blocked GNPS.** In the constrained version of the blocked GNPS, since the number of distinct q-vector types $M$ (e.g., $M = 7$ when examining three independent attributes) typically exceeds the block size $J_s$, only one item from each q-vector type is required. The items with the highest discrimination power (defined in Eq 3) within each q-vector type are identified, and the top $J_s$ items from this subset are selected to form the block.

In the simulation study, we compared the constrained version with the unconstrained one, assuming that similar to the case of parametric CD-CAT, the constrained version would perform better under some conditions. By doing so, we aimed to identify which version offers higher classification accuracy, thereby providing insights for practitioners seeking to implement blocked-design CAT in diverse testing contexts.

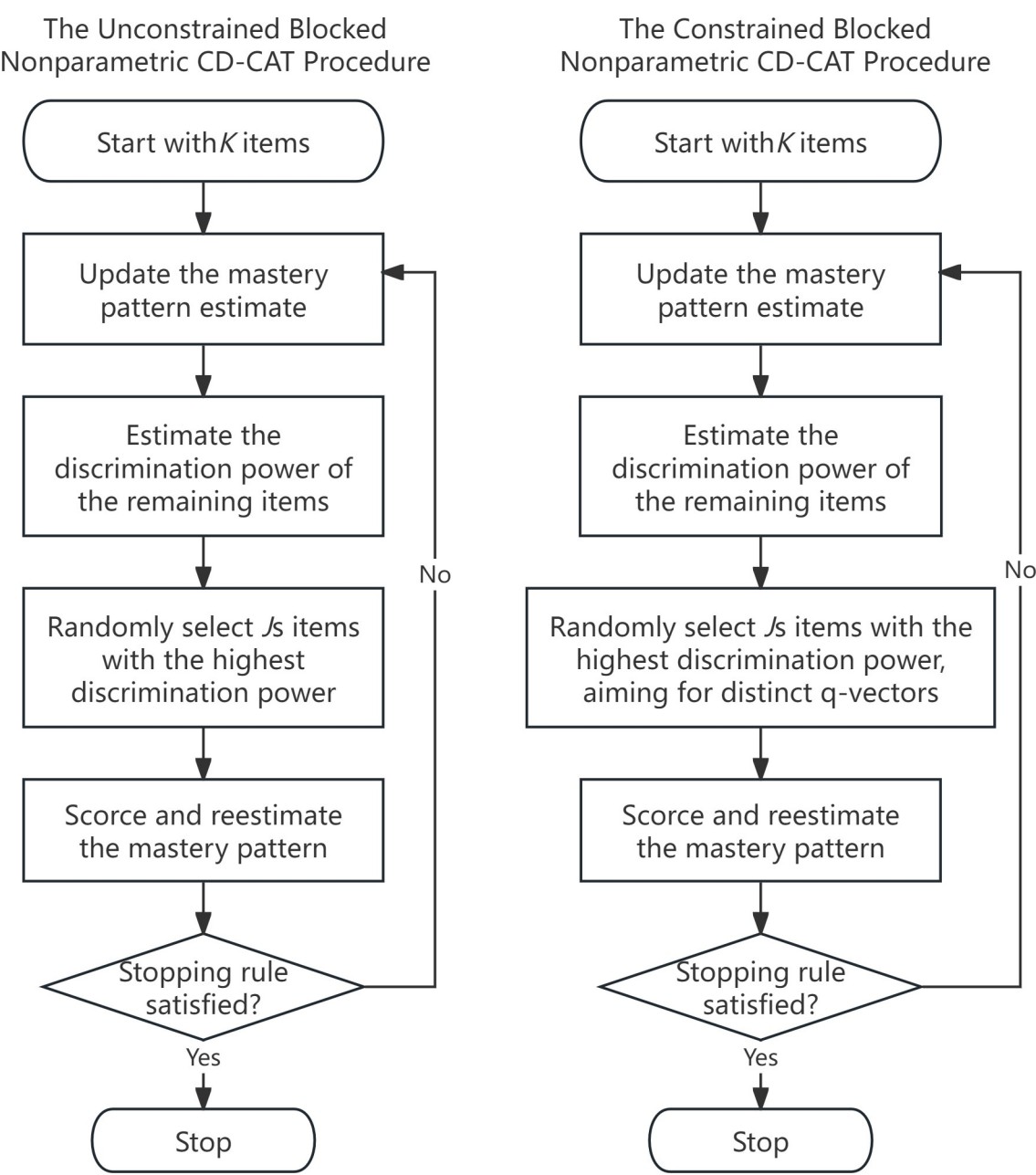

**Fig 1. The two versions of the blocked nonparametric CD-CAT procedure.** CD-CAT = cognitive diagnosis computerized adaptive testing.

### Aims of the simulation study

A simulation study was conducted to assess the performance of the proposed blocked procedure. Specifically, the study sought to address the following question: How does the classification accuracy of nonparametric blocked CD-CAT compare to that of parametric blocked CD-CAT, both with accurate and inaccurate item parameters, under various conditions? These conditions included block size, constraints on q-vectors within blocks, test length, item quality, and the data generation model.

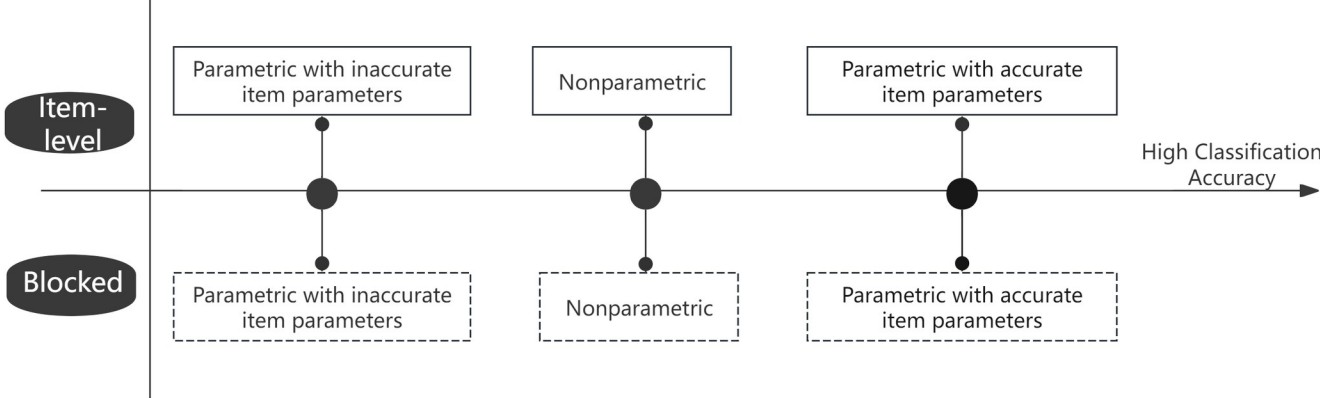

**Fig 2. Schematic representation of parametric vs. nonparametric CD-CAT methods for classification accuracy.**

## Method

### Simulation design

This study compared classification accuracy between nonparametric and parametric blocked CD-CAT. The nonparametric item selection strategies are the blocked NPS and blocked GNPS, as proposed in the previous section. These procedures were compared to the parametric methods, specifically the blocked versions of PWKL and MPWKL [12]. All simulations were implemented using self-developed code in R 4.1.3.

Fig 2 presents a schematic comparison between parametric and nonparametric methods. Previous research often incorporates an item calibration process to simulate conditions with limited calibration data [9, 11]. However, this approach frequently introduces significant error due to small sample sizes, leading to inaccurate item parameters and a corresponding reduction in the classification accuracy of parametric methods. As a result, nonparametric methods often achieve higher classification accuracy in scenarios with limited calibration data [9, 11]. Similarly, we expect the nonparametric blocked CD-CAT to outperform its parametric counterpart under such conditions, as shown in Fig 2. Nonetheless, parametric methods would rarely be applied in real-world situations with limited calibration data, rendering this comparison less relevant. In contrast, research on parametric methods typically assumes true item parameters in simulations [12]. Accordingly, this study uses parametric methods with true item parameters as a benchmark, representing the upper bound of accuracy. If the performance gap between nonparametric blocked CD-CAT and this high standard is minimal, the proposed method can be considered to demonstrate strong classification performance.

This study also investigated the influence of several key factors on the classification accuracy of blocked CD-CAT. Two primary factors were examined: block size (with three levels) and the application of constraints to the q-vectors within blocks (with two levels). The selection of the three block sizes was informed by previous research on parametric blocked CD-CAT, where a block size of 1 corresponds to item-level CD-CAT [12]. Both the blocked NPS and blocked GNPS methods were tested in two versions, one with constraints on the q-vectors within blocks (C) and one without constraints (NC).

Additional factors examined in this study include the data generation model (two levels), test length (three levels), and item quality (two levels), as summarized in Table 1. The DINA model and the additive cognitive diagnosis model (A-CDM) were used for data generation and parametric CD-CAT simulations, as their underlying assumptions align well with cognitive processes commonly observed in academic contexts, such as mathematics education [32].

**Table 1. Factors and levels of the simulation study.**

| Factor | Levels |
|---|---|
| Item Selection Strategy/Mastery Pattern Estimation Method | Parametric: blocked PWKL/MLE, MPWKL/MLE |
| | Nonparametric: blocked NPS/NPC, blocked GNPS/GNPC |
| Version of the Block Design | UC, C |
| Block Size ($J_s$) | 1, 2, 4 |
| Data Generation Model | DINA, A-CDM |
| Test Length ($J$) | 17, 25, 33 |
| Item Quality | High (U(0, 0.15)), Low (U(0.15, 0.3)) |

Note. UC = unconstrained; C = constrained version; DINA = deterministic input noisy "and" gate; A-CDM = additive cognitive diagnosis model; PWKL = posterior-weighted K-L index; MPWKL = modified posterior-weighted K-L index; NPS = nonparametric classification item selection strategy; GPNS = general nonparametric classification item selection strategy.

The DINA model assumes a conjunctive relationship between attributes, consistent with the assumptions of the NPS method. In contrast, the A-CDM adopts a compensatory relationship between attributes, which is compatible with the GNPS method [8–11, 27, 32]. Parametric methods with inaccurately estimated item parameters were only used in conjunction with the DINA model for data generation, as this was not the primary focus of the study.

At the initial stage, five items were selected using the Q-optimal algorithm [33], after which the blocked design was implemented. Test lengths of 17, 25, and 33 items were used to represent short, medium, and long tests, respectively. A test length of 17 implies that 12 additional items were administered after the initial five, corresponding to three blocks when the block size is four. These test lengths exceed those typically employed in parametric blocked designs [12], as nonparametric methods generally require longer tests to achieve comparable efficiency.

The selection of item quality levels was informed by previous research [9–11, 27]. A detailed explanation of how item banks with varying quality levels were generated is provided in subsequent sections.

## Item selection strategies

Conventional parametric (e.g., PWKL and MPWKL) and nonparametric (e.g., NPS and GNPS) item selection strategies were employed when the block size was 1 [2, 9, 11, 31]. For block sizes greater than 1, the proposed blocked procedures—blocked NPS and blocked GNPS —were utilized, with parametric blocked procedures included for comparison. The R package GDINA was used for item calibration in the parametric item selection strategies, specifically for the DINA and A-CDM models [34, 35], using the marginal maximum likelihood estimation (MMLE) method with the Expectation-Maximization (EM) algorithm.

## Mastery pattern estimation

Parametric CD-CAT methods estimated examinees' mastery patterns based on the DINA or A-CDM using maximum likelihood estimation (MLE) with either the true or estimated item parameters [7]. Nonparametric CD-CAT methods, including conventional NPS and GPNS as well as their blocked versions, employed NPC or GNPC for estimating examinees' mastery patterns [8, 10]. The weights in GNPC were estimated using the calibration sample [10].

## Item pool generation

Following previous research, the tests assess five attributes (K) [11, 36], resulting in a total of 31 possible item types. The item pool size (J) is set to 1240, with 40 items per type. The large item pool serves to ensure that the proposed method's efficacy is assessed independently of the item pool's size, allowing for a clearer evaluation of its potential under optimal conditions.

Each item in the pool is characterized by parameters such as guessing and slipping, which indicate item quality. Higher quality items exhibit greater discrimination. Guessing parameters ($P(Y_j = 1|\alpha = 0)$) and slipping parameters ($P(Y_j = 0|\alpha = 1)$) for high-quality items are uniformly distributed between 0 and 0.15, whereas those for low-quality items are uniformly distributed between 0.15 and 0.3.

## Examinee generation

This study involves two approaches to generating examinee response data. The first approach involves simulating examinees for the CD-CAT, adhering to the efficient methodology proposed by Kaplan et al. [31]. Instead of simulating examinees cross all possible attribute profiles, we generated examinees corresponding to six selected attribute profiles, representing key stages in attribute mastery. These profiles range from no mastery to full mastery: $\alpha_0$ = (0,0,0,0,0), $\alpha_1$ = (1,0,0,0,0), $\alpha_2$ = (1,1,0,0,0), $\alpha_3$ = (1,1,1,0,0), $\alpha_4$ = (1,1,1,1,0), and $\alpha_5$ = (1,1,1,1,1), with 200 examinees assigned to each profile. Given that the attributes are assumed to be independent in this study, all attribute mastery patterns involving the same number of mastered attributes are considered equivalent in terms of their simulation outcomes.

The second approach was used to generate calibration samples for the blocked GNPS to determine the weights. A calibration sample is also needed for the parametric methods using item parameter estimates instead of the true values. Specifically, the attribute profiles of the examinees in the calibration samples were generated using the multivariate normal threshold model with a covariance of 0.5 [7]. To demonstrate the impact of small calibration samples on the performance of the parametric methods, the calibration sample size ($N_0$) was set to 100 [9, 11].

## Evaluation criterion

The evaluation criterion chosen for this study is the conditional classification accuracy, which computes the conditional pattern classification accuracy for six mastery patterns: $\alpha_0$ = (0,0,0,0,0), $\alpha_1$ = (1,0,0,0,0), $\alpha_2$ = (1,1,0,0,0), $\alpha_3$ = (1,1,1,0,0), $\alpha_4$ = (1,1,1,1,0), $\alpha_5$ = (1,1,1,1,1) [29]. The conditional pattern correct classification rate of $\alpha_l$ is expressed as

$$PCCR_l = \frac{\sum_{i=1}^{N} I(\alpha_{il} = \hat{\alpha}_{il})}{N},$$ (4)

where $I$ is the indicator function, $N$ represents the number of examinees, and $l$ is the index of attribute mastery patterns.

Assuming that the mastery pattern of the examinee population follows a uniform distribution, the marginal pattern classification accuracy for each method can be estimated by weighting these six patterns. The weights are 1/32, 5/32, 10/32, 10/32, 5/32, and 1/32, representing the proportions of mastery patterns with 0, 1, 2, 3, 4, and 5 attributes mastered out of 32 possible patterns [31].

Based on prior research, differences in classification accuracy are considered meaningful when equal to or greater than 0.05; differences greater than or equal to 0.03 but less than 0.05 are considered trivial, while differences less than 0.03 are deemed negligible [12, 31].

**Table 2. Pattern correct classification rates of parametric and non-parametric CD-CAT using the DINA model.**

| Item quality | $J$ | Js | PWKL [a] | | MPWKL [a] | | PWKL-EST [b] | | MPWKL-EST [b] | | NPS | | GNPS | |
|---|---|---|---|---|---|---|---|---|---|---|---|---|---|---|
| | | | UC | C | UC | C | UC | C | UC | C | UC | C | UC | C |
| Low | 17 | 1 | 0.779 | | 0.820 | | 0.575 | | 0.574 | | 0.650 | | 0.396 | |
| | | 2 | 0.767 | 0.771 | 0.792 | 0.800 | 0.535 | 0.563 | 0.526 | 0.564 | 0.621 | 0.635 | 0.399 | 0.407 |
| | | 4 | 0.643 | 0.733 | 0.695 | 0.757 | 0.471 | 0.518 | 0.528 | 0.540 | 0.543 | 0.558 | 0.335 | 0.325 |
| | 25 | 1 | 0.922 | | 0.922 | | 0.702 | | 0.717 | | 0.803 | | 0.548 | |
| | | 2 | 0.904 | 0.918 | 0.928 | 0.925 | 0.687 | 0.721 | 0.693 | 0.713 | 0.773 | 0.785 | 0.541 | 0.564 |
| | | 4 | 0.815 | 0.886 | 0.888 | 0.893 | 0.652 | 0.661 | 0.686 | 0.666 | 0.737 | 0.721 | 0.457 | 0.487 |
| | 33 | 1 | 0.970 | | 0.973 | | 0.771 | | 0.775 | | 0.904 | | 0.671 | |
| | | 2 | 0.961 | 0.957 | 0.979 | 0.980 | 0.765 | 0.767 | 0.766 | 0.764 | 0.886 | 0.900 | 0.665 | 0.645 |
| | | 4 | 0.913 | 0.950 | 0.959 | 0.952 | 0.740 | 0.743 | 0.762 | 0.737 | 0.857 | 0.872 | 0.564 | 0.625 |
| High | 17 | 1 | 1.000 | | 1.000 | | 1.000 | | 1.000 | | 0.988 | | 0.922 | |
| | | 2 | 0.998 | 0.998 | 1.000 | 0.998 | 0.998 | 1.000 | 0.999 | 1.000 | 0.980 | 0.978 | 0.841 | 0.864 |
| | | 4 | 0.984 | 0.994 | 0.986 | 1.000 | 0.885 | 0.997 | 0.999 | 0.995 | 0.909 | 0.915 | 0.764 | 0.785 |
| | 25 | 1 | 1.000 | | 1.000 | | 1.000 | | 1.000 | | 0.988 | | 0.964 | |
| | | 2 | 1.000 | 1.000 | 1.000 | 1.000 | 1.000 | 1.000 | 1.000 | 1.000 | 0.999 | 0.998 | 0.926 | 0.940 |
| | | 4 | 1.000 | 1.000 | 0.998 | 1.000 | 0.999 | 1.000 | 1.000 | 0.999 | 0.984 | 0.994 | 0.876 | 0.880 |
| | 33 | 1 | 1.000 | | 1.000 | | 1.000 | | 1.000 | | 1.000 | | 0.983 | |
| | | 2 | 1.000 | 1.000 | 1.000 | 1.000 | 1.000 | 1.000 | 1.000 | 1.000 | 1.000 | 1.000 | 0.972 | 0.975 |
| | | 4 | 1.000 | 1.000 | 1.000 | 1.000 | 1.000 | 1.000 | 1.000 | 1.000 | 0.998 | 0.998 | 0.948 | 0.956 |

Note. UC = unconstrained; C = constrained version; DINA = deterministic input noisy "and" gate; J = test length; Js = block size; PWKL = posterior-weighted K-L index; MPWKL = modified posterior-weighted K-L index; NPS = nonparametric classification item selection strategy; GPNS = general nonparametric classification item selection strategy.

[a] True item parameters were used for item selection and attribute profile estimation.

[b] Item parameter estimates with a calibration sample of 100 were used for item selection and attribute profile estimation.

## Results

Initially, we compared nonparametric blocked CD-CAT procedures with parametric CD-CAT regarding pattern classification accuracy when response data were generated based on the DINA model. Employing true item parameters in the parametric approach is considered ideal, which expectedly resulted in superior classification accuracy compared to the nonparametric method proposed in this study. As shown in Table 2, across various conditions of item pool quality and test length, the item-level parametric CD-CAT (i.e., block size of 1) achieved the highest classification accuracy. If the classification accuracy of the nonparametric method approximates this ideal scenario, this suggests the efficacy of the nonparametric approach.

A comparison between nonparametric and parametric approaches indicates that in scenarios with a low-quality item pool, the nonparametric method performed worse than the parametric counterpart; however, in scenarios with a high-quality item pool, the nonparametric method with NPS approached the performance of the parametric method using true item parameters. Given that the data generation model is DINA, NPS generally outperformed GNPS in most conditions. Chiu and Chang suggest that a pattern classification accuracy of 0.8 is deemed satisfactory [11]. Regarding NPS, pattern classification accuracy typically met this criterion except in scenarios with low item quality and short test length. With GNPS, pattern classification accuracy was predominantly satisfactory in high-quality item scenarios.

In general, the blocked design resulted in lower classification accuracy compared to item-level CD-CAT (i.e., block size of 1), with classification accuracy decreasing as the block size

increased, across both parametric and nonparametric methods. Yet, with a block size of 2, the classification accuracy was only marginally lower than that of item-level CD-CAT. The impact of module size diminished when item quality was high and test length was moderate to long.

The impact of constraining the q-vector of blocked items on classification accuracy varied across different conditions. In scenarios with high item quality, constraining the q-vector of blocked items had minimal effect on enhancing pattern classification accuracy. In scenarios with low item quality, constraining the q-vector of blocked items could enhance pattern classification accuracy with both parametric item selection strategies, particularly when the test length was short and the blocks were large. Regarding the nonparametric item selection strategy GNPS, constraining the q-vector of blocked items could improve pattern classification accuracy when item quality was low, test length was moderate or long, and the blocks were large. Nevertheless, irrespective of item quality, test length, or block size, constraining the q-vector of blocked items did not significantly improve the pattern classification accuracy of NPS.

Furthermore, we observed that in scenarios with low item quality and short test length, MPWKL demonstrated higher classification accuracy compared to PWKL, aligning with the results reported by Kaplan and de la Torre [10]. In both parametric and nonparametric CD-CAT, while controlling for other experimental conditions, the classification accuracy was higher when using high-quality item pools compared to low-quality item pools, and it increased with test length, in line with previous studies [11, 12, 19, 31]. With GNPS, enhancing item quality had a greater impact on improving pattern classification accuracy, particularly in scenarios with short test length.

We employed the methodology from previous studies, calibrating parameters using small samples for parametric methods, and utilized the DINA model as the data generation model as an example. Parameter estimates obtained from small calibration samples are expected to contain notable errors. This approach is employed to showcase the superiority of nonparametric blocked CD-CAT over parametric methods in scenarios with limited access to calibration samples.

The findings (see Table 2) align with previous research on nonparametric methods: in scenarios with limited calibration samples, the classification accuracy of parametric methods was notably lower compared to using true parameters, whereas nonparametric blocked CD-CAT attained high accuracy without necessitating large calibration samples, especially in cases of low item quality.

Table 3 shows the performances of the item selection strategies when the response data were generated using the A-CDM, which assumes a compensatory relationship between attributes. We observed relatively low overall pattern classification accuracy for the NPS method, which assumes a conjunctive relationship between attributes. Overall, the effects of item quality, test length, and block size on pattern classification accuracy were generally comparable to those observed when response data were generated from the DINA model, which shares the same assumption as the NPS method.

An examination of the impact of constraining the q-vector of blocked items revealed that in scenarios of low item quality, for GNPS, with moderate test length and large blocks, constraining the q-vector of blocked items resulted in a 6.7% increase in pattern classification accuracy. In scenarios characterized by high item quality and short test length, constraining the q-vector of blocked items led to an approximately 7% increase in classification accuracy, irrespective of block size.

In summary, when comparing the results obtained from response data generated using the DINA model versus the A-CDM model, the blocked CD-CAT with NPS achieved better classification accuracy when response data are generated from the DINA model, whereas the

**Table 3. Pattern correct classification rates of parametric and non-parametric CD-CAT using the A-CDM.**

| Item quality | $J$ | $J_s$ | PWKL[a] | | MPWKL[a] | | NPS | | GNPS | |
|---|---|---|---|---|---|---|---|---|---|---|
| | | | UC | C | UC | C | UC | C | UC | C |
| LQ | 17 | 1 | 0.795 | | 0.784 | | 0.429 | | 0.473 | |
| | | 2 | 0.771 | 0.724 | 0.760 | 0.744 | 0.432 | 0.402 | 0.461 | 0.450 |
| | | 4 | 0.722 | 0.681 | 0.706 | 0.682 | 0.335 | 0.406 | 0.391 | 0.403 |
| | 25 | 1 | 0.901 | | 0.901 | | 0.545 | | 0.593 | |
| | | 2 | 0.888 | 0.884 | 0.885 | 0.883 | 0.525 | 0.502 | 0.600 | 0.593 |
| | | 4 | 0.866 | 0.852 | 0.868 | 0.836 | 0.435 | 0.503 | 0.525 | 0.560 |
| | 33 | 1 | 0.960 | | 0.954 | | 0.611 | | 0.699 | |
| | | 2 | 0.955 | 0.947 | 0.953 | 0.947 | 0.593 | 0.572 | 0.681 | 0.701 |
| | | 4 | 0.950 | 0.925 | 0.953 | 0.897 | 0.521 | 0.589 | 0.649 | 0.656 |
| HQ | 17 | 1 | 0.998 | | 1.000 | | 0.720 | | 0.846 | |
| | | 2 | 0.998 | 1.000 | 1.000 | 0.995 | 0.711 | 0.751 | 0.772 | 0.819 |
| | | 4 | 0.978 | 0.990 | 0.968 | 0.993 | 0.740 | 0.710 | 0.771 | 0.826 |
| | 25 | 1 | 1.000 | | 1.000 | | 0.754 | | 0.916 | |
| | | 2 | 1.000 | 1.000 | 1.000 | 1.000 | 0.761 | 0.793 | 0.852 | 0.872 |
| | | 4 | 0.999 | 0.998 | 0.998 | 1.000 | 0.783 | 0.768 | 0.845 | 0.855 |
| | 33 | 1 | 1.000 | | 1.000 | | 0.797 | | 0.930 | |
| | | 2 | 1.000 | 1.000 | 1.000 | 1.000 | 0.812 | 0.841 | 0.904 | 0.923 |
| | | 4 | 1.000 | 1.000 | 1.000 | 1.000 | 0.823 | 0.816 | 0.908 | 0.910 |

Note. A-CDM = additive cognitive diagnosis models; J = test length; Js = block size; PWKL = posterior-weighted K-L index; MPWKL = modified posterior-weighted K-L index; NPS = nonparametric classification item selection strategy; GPNS = general nonparametric classification item selection strategy; UC = unconstrained; C = constrained version.

[a] True item parameters were used for item selection and attribute profile estimation.

blocked procedure with GNPS demonstrated relatively higher classification accuracy when response data were generated from the A-CDM model. Both methods achieved classification accuracy close to the ideal conditions of parametric methods or exceeding the standard of 0.8 when item quality was high or the test length was sufficiently long.

## Discussion

This study presents a blocked nonparametric CD-CAT procedure that integrates a nonparametric item selection strategy with a multi-stage online test assembly. Extending the nonparametric item selection strategies (NPS and GNPS) to the blocked versions, this procedure enables item review within the block. Simulation studies examined the impact of various factors on the classification accuracy rate of the blocked nonparametric CD-CAT.

Previous research on item-level CD-CAT has consistently indicates that NPS and GNPS outperform parametric methods when the calibration sample is small ($N_0 < 1000$), largely due to significant estimation errors in item parameters used by parametric methods [11, 27]. The observed results align with theoretical expectations: NPS operates independently of a calibration sample, GNPS remains effective even with a small calibration sample, whereas parametric methods require a considerably larger calibration sample to achieve reliable estimates. Our findings further corroborate the advantages of non-parametric approaches in CD-CAT, as demonstrated by the proposed blocked procedure.

As a complement to previous studies, our study focused on comparing the non-parametric CD-CAT with the parametric CD-CAT using true item parameters in item selection and

attribute profile estimation. In contrast, previous studies primarily chose calibration samples much smaller than optimal for parametric methods [11, 27]. By comparing non-parametric methods with parametric CD-CAT utilizing true item parameters, our study highlights the disparity between non-parametric approaches and the ideal scenario represented by parametric methods. This disparity diminishes with longer tests and higher item bank quality. Notably, when response data is generated based on the DINA model and the test length is moderate or longer, the gap between blocked NPS and the ideal scenario becomes minimal. This finding is of particular significance for practitioners, as the non-parametric method offers a considerably simpler and more cost-effective alternative to parametric methods, which require large calibration samples to achieve optimal performance.

The proposed methods demonstrate satisfactory pattern classification accuracy under specific conditions when evaluated using an absolute criterion. Chiu and Chang's research defines a classification accuracy of 0.8 as satisfactory [11]. Our simulation studies demonstrated that under the DINA model, both blocked NPS and GNPS achieved a classification accuracy of 0.8 when item quality was high. Additionally, the blocked NPS model reached the satisfactory classification rate of 0.8 even when item quality was low, provided that the test length was sufficiently long. On the other hand, the A-CDM model showed that blocked GNPS achieved a satisfactory accuracy level of 0.8 with a high-quality item pool, and the blocked NPS reached the same level of accuracy when both the test length was extended and the item quality was high.

The evaluation of the effects of block size and the constraint of q-vectors within a block should take into account the quality of the item pool and the test length. High-quality item pools resulted in higher classification accuracy rates compared to low-quality item pools, and accuracy rates increased with longer tests, consistent with previous research [11, 12, 19, 31]. As the block size increases, the test becomes less adaptive and the classification accuracy generally decreased akin to the difference between Multi-Stage Testing (MST) and item-level CAT [12]. This decline in accuracy was mitigated by enhancements in item quality and test length. Notably, when the block size was set to 2, the classification accuracy rate did not significantly differ from that of item-level non-parametric CD-CAT. Similarly, a block size of 4 showed no significant difference in accuracy under conditions of high-quality items and longer test lengths. Additionally, under certain conditions—such as when response data are generated based on the A-CDM, with a moderate test length and large block size—constraining the q-vector within blocked items can increase classification accuracy in the GNPS.

Integrating block design with nonparametric CD-CAT offers additional advantages. The block design reduces testing time by estimating the attribute mastery pattern less frequently. A small block size does not significantly lower the classification accuracy compared to item-level testing. More importantly, the block design allows examinees to review and modify their answers within each block, reducing test anxiety and minimizing errors. Unlike Stocking I, which offers item review only at the end [15], the block-level item review design immediately applies modified answers to subsequent item selection, enhancing final classification accuracy. Ultimately, the choice of block size involves a trade-off between adaptivity and other practical considerations, such as allowing students to review and change their answers.

The choice between the blocked NPS and blocked GNPS methods depends on the nature of the relationships among attributes. When the assumptions of the DINA or DINO models apply—where a conjunctive or disjunctive relationship is likely—the blocked NPS method is recommended. For tests with low-quality items, a longer test length is necessary to achieve accurate classifications, whereas moderate test lengths are sufficient when item quality is high, with block sizes of 2 or 4 providing acceptable classification accuracy. Conversely, if the

attributes exhibit a more complex relationship, as in the case of the A-CDM, the blocked GNPS procedure is more suitable. Importantly, the exact CDM need not be known, as the weights in the GNPS method automatically adjust to the complexity and variability of the response data [10, 11].

High item quality and longer test lengths contribute to strong classification accuracy for blocked GNPS, with both block sizes of 2 and 4 performing well. Ultimately, the choice of block size involves a trade-off between adaptivity and other practical considerations, such as allowing students to review and change their answers. Practitioners must determine the tolerable level of loss in classification accuracy when deciding on the appropriate block size for their specific context [12]. The constrained version of the blocked procedure is also worth considering, as it can maintain or even enhance classification accuracy under certain conditions by constraining the q-vectors of blocked items. Since items with identical q-vectors assess the same content or skills, incorporating a variety of item types within a block is beneficial to avoid redundancy and reduce examinee boredom.

Concerns may arise regarding item quality when employing the proposed non-parametric approach, particularly in constructing the item pool without relying on a parametric model. Compared to parametric approaches, non-parametric CD-CAT imposes relatively lower demands on item quality. In non-parametric methods, the reliance on strict model assumptions is reduced, which allows for greater flexibility in item selection. As a result, non-parametric methods can effectively function even when items exhibit lower levels of discrimination. If high-quality items are difficult to obtain, non-parametric methods may be more cost-effective than their parametric counterpart.

Nevertheless, it is important to emphasize that the fundamental criteria of item quality remain crucial, regardless of whether a parametric or non-parametric model is used when constructing the item pool. For example, key aspects such as content validity and the clarity of item wording are universally applicable. To construct the item pool without relying on a specific parametric model, we suggest focusing on general principles of test development. This includes ensuring that the items comprehensively cover the targeted skills or attributes and that they are of appropriate difficulty for the population being tested. Furthermore, empirical analyses, such as item-total correlations, can be employed to identify and remove poorly functioning items. These steps help in maintaining the overall quality of the item bank, even in the absence of model-based item calibration.

The proposed nonparametric blocked CD-CAT has demonstrated expected performance and presents a promising tool for classroom assessment. However, this study is not without its limitations, underscoring the need for further research. First, the nonparametric approach was introduced to address challenges associated with parametric methods, particularly in scenarios involving small calibration samples. However, recent advancements in parametric methods have yielded techniques that enhance estimation accuracy, even with small sample sizes ($\leq$100) [37]. Future research should incorporate these developments when comparing non-parametric and parametric approaches. Second, our simulation study focused on evaluating the proposed method using a large item pool to minimize the confounding effects of pool size. However, since large item pools may be impractical in real-world settings, future research should assess its performance with smaller pools and determine the minimum pool size required. Third, the simulation study focused on data generated from a single model (either DINA or A-CDM). Future research should explore the performance of the proposed procedures in item pools where various CDMs are mixed. Based on previous findings with item-level CD-CAT using GNPS [11], we anticipate that the blocked GNPS method would outperform the blocked NPS in such scenarios.

## Author Contributions

**Conceptualization:** Yuqing Yuan, Feng Li.

**Data curation:** Yuqing Yuan, Ziying Qiu.

**Formal analysis:** Jiahui Zhang, Ziying Qiu.

**Methodology:** Feng Li.

**Supervision:** Feng Li.

**Writing – original draft:** Jiahui Zhang, Ziying Qiu.

**Writing – review & editing:** Jiahui Zhang, Yuqing Yuan, Feng Li.

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
