## [Decision Letter · Decision Letter 0]

18 Jul 2024

PONE-D-24-21730Introducing a Blocked Procedure in Nonparametric CD-CATPLOS ONE

Dear Dr. Li,

Thank you for submitting your manuscript to PLOS ONE. After careful consideration, we feel that it has merit but does not fully meet PLOS ONE’s publication criteria as it currently stands. Therefore, we invite you to submit a revised version of the manuscript that addresses the points raised during the review process.

1. Respond to reviewers' requests on a case-by-case basis and reflect them in revised manuscript.

2. Respond to one of my additional question.

We look forward to receiving your revised manuscript.

Kind regards,

Peida Zhan

Academic Editor

PLOS ONE

Journal Requirements:

3. In the online submission form, you indicated that "The data underlying the results presented in the study are available from the correpondence author."

Additional Editor Comments:

After the review by two professional reviewers and my personal reading, I think that this study has a certain degree of innovation on the topic of nonparametric CDCAT. On the whole, I agree with the comments of the two reviewers; meanwhile, I would like to add one question that the authors would like to add in the revised manuscript: the problem of item bank construction of non-parametric CDCAT, i.e., does non-parametric CDCAT need to consider the quality of the items in the construction of item banks, and if it needs to consider the quality of the items, how to construct a high-quality item bank without using a model?

Reviewers' comments:

Reviewer's Responses to Questions

**Comments to the Author**

1. Is the manuscript technically sound, and do the data support the conclusions?

Reviewer #1: Yes

Reviewer #2: Yes

2. Has the statistical analysis been performed appropriately and rigorously? 

Reviewer #1: Yes

Reviewer #2: Yes

3. Have the authors made all data underlying the findings in their manuscript fully available?

Reviewer #1: Yes

Reviewer #2: Yes

4. Is the manuscript presented in an intelligible fashion and written in standard English?

Reviewer #1: Yes

Reviewer #2: No

5. Review Comments to the Author

Reviewer #1: 1. The authors explored two versions of each blocked procedure in the simulation study, can the authors explain the reasons?

2. Can the authors list the detailed steps of the blocked NPS and GNPS methods in the main text?

3. how to determine the number of items that each q-vector type needed in the first step for the constrained blocked GNPS (lines 208-210)

4. The reason of the simulation design should be explained.

5. During the data generation, only six attribute profiles were generated, the rests were discarded, the authors should explain this setting.

6. The authors should explain the purpose and reason of constraining the q-vector of blocked items. Did this situation (i.e., only those items with different q-vectors would be selected into the block) would be occurred in empirical study?

7. During the simulation two (lines 375-385), which methods (package(s)) were used to calibrate item parameters? The original EM algorithm? or the Bayes Modal Estimation method (Ma & Jiang, 2021). Meanwhile, how many people were used to calibrate the items?

Ma, W., & Jiang, Z. (2021). Estimating cognitive diagnosis models in small samples: Bayes modal estimation and monotonic constraints. Applied Psychological Measurement, 45(2), 95-111.

8. During the calibration of item parameters, how to calibrate the item parameters? all 1240 items were assembled to create a single test, then calibrated using small samples (e.g., 100)? Is that realistic? Can the authors adopted the matrix sampling technique to calibrate the item parameters?

9. Can the authors explain why the classification accuracy rate decreased as the block size increased?

10. The authors information were missing for the 18th reference in the "references" section

Reviewer #2: This study intends to incorporate a blocked procedure into nonparametric CD-CAT to allow item review and answer modification while maintaining classification accuracy. The results showed that the blocked procedure with increasing the block size reduced pattern classification; however, this reduction decreased as item quality and test length increased. This study is beneficial for implication; however, the following particular questions must be addressed in order for merit to be published in PLOS ONE.

1. The simulation design used a parametric CD-CAT design, with some uncommon settings. Can the authors explain on the rationale for doing such a design, as well as the relationship to a non-parametric CD-CAT and more explanations of the simulation design? The explanations and specifications can assist the reader in understanding the distinction between non-parametric and parametric CD-CATs, conducting a non-parametric CD-CAT with proper simulation settings, and drawing the simulation results or implications. Such as, a big item pool (1240 items) is employed, which is rare in small-scale or classroom contexts. Was the sample size of 200 subjects for each studied attribute mastery pattern large enough to make an inference? Why were just the DNIA and A-CDM models used? The rationale for setting up the three levels of test lengths, etc.

2. The authors recommended that the blocked NPS is more suited for the DINA model and the blocked GNPS for the A-CDM model. The empirical data may be collected from different types of CDMs; could the authors provide further explanations / advices on when and how to utilize a suitable item selection method for a non-parametric CD-CAT when various CDMs are used. For example, when using a non-parametric CD-CAT with the DINO model, which item selection strategy should be used?

3. The results of the non-parametric CD-CAT were not included in Table 3.

4. On pages 24 and 29, the authors mentioned that “The violation of NPS's underlying assumptions when generating response data from the A-CDM model resulted in relatively low overall pattern classification accuracy for NPS.” and “When the response data better conform to the A-CDM model assumptions, using GNPS is more effective.” It is hard to follow if a reader is not familiar with NPS/GNPS and A-CDM.

5. Lines 451-454 on page 28 need to be reorganized. Based on Tables 2 and 4, the DINA model produced a classification rate of .8 when the item quality was high for both blocked NPS and GNPS, as well as when the item quality was low and the test length was lengthy for the blocked NPS. While the A-CDM model yielded a classification rate of .8 with the high-quality item pool for the blocked GNPS and a high item quality with the longer test for the blocked NPS.

6. Before submitting the manuscript, please check for citations (e.g., introducing Q-matrix and attribute mastery pattern), grammatical errors, and coherence (e.g., missing notations for Equation 3, the simulation data generated by the DINA and A-CDM models but GNPS used the weights based on the DINA and DINO models on page 17).

6. PLOS authors have the option to publish the peer review history of their article (what does this mean?). If published, this will include your full peer review and any attached files.

Reviewer #1: No

Reviewer #2: **Yes: **Chia-Ling Hsu

---

## [Author Response · Author response to Decision Letter 0]

2 Oct 2024

Dear Editor,

We sincerely appreciate the time and effort you and the reviewers have invested in evaluating our manuscript. We have carefully considered all the valuable comments and suggestions provided, and have made comprehensive revisions to the paper. Below is a summary of the major modifications:

Detailed steps and the rationale behind the proposed procedures have been added.

The Methods section has been expanded to clarify the rationale for the simulation plan.

The tables in the Results section have been reorganized to improve clarity.

The Discussion section has been strengthened to address the reviewers’ concerns more thoroughly.

We have conducted a comprehensive grammar check throughout the manuscript. All identified grammatical errors have been corrected to ensure that the text is clear, concise, and professional.

We would like to thank you for your question regarding the construction of item banks in non-parametric CD-CAT. We appreciate the opportunity to clarify this aspect of our study.

Compared to parametric approaches, non-parametric CD-CAT indeed places relatively lower demands on item quality. In non-parametric methods, the reliance on strict model assumptions is reduced, which allows for greater flexibility in item selection. As a result, non-parametric methods can effectively function even when items exhibit lower levels of discrimination.

However, it is important to emphasize that the fundamental criteria of item quality remain crucial, regardless of whether a parametric or non-parametric model is used. For example, key aspects such as content validity and the clarity of item wording are universally applicable. To construct a high-quality item bank without relying on a specific measurement model, we suggest focusing on general principles of test development. This includes ensuring that the items comprehensively cover the targeted skills or attributes and that they are of appropriate difficulty for the population being tested. Furthermore, empirical analyses, such as item-total correlations, can be employed to identify and remove poorly functioning items. These steps help in maintaining the overall quality of the item bank, even in the absence of model-based item calibration.

In response to your comment, we have revised and supplemented the relevant section of the Discussion in our manuscript (see pp. 36-37). We have expanded our discussion on item bank construction, addressing the specific considerations for non-parametric CD-CAT. This includes the challenges of maintaining item quality and the strategies for constructing a robust item bank without relying on parametric models.

We hope that this response and the corresponding revisions to the manuscript address your concerns effectively.

Please find attached our detailed responses to the reviewers' comments, along with an outline of the revisions made. All modiﬁcations are highlighted in the revised manuscript. We greatly appreciate your continued consideration of our work and the constructive feedback provided.

Sincerely,

Authors

Response to Reviewer 1

We appreciate the time and eﬀort that you have devoted to giving us insightful suggestions and comments. We thoroughly revised our paper according to the suggestions and comments from you and the other reviewer. 

Below, we have arranged our responses below in the same sequence as your original comments/suggestions and outline changes made to the paper according to your comments/suggestions. All modiﬁcations are highlighted in the revised manuscript.

1.The authors explored two versions of each blocked procedure in the simulation study, can the authors explain the reasons?

Reply: 

Our decision to explore multiple versions of the blocked procedure is grounded in the work of Kaplan and de la Torre (2020), who proposed different versions of the blocked parametric CD-CAT, including unconstrained, constrained, and hybrid approaches. 

We provided a detailed explanation of the rationale behind the constrained version in the revised manuscript (pp. 13-14): 

“The constrained version of the blocked procedure aims to make the q-vectors of items within the same block as distinct as possible. The rationale behind introducing the constrained version lies in its potential improvement on the performance of the adaptive testing procedure. Previous research in parametric methods indicated that administering items with the same q-vector repeatedly did not provide additional information for improving measurement precision [29]. Kaplan and de la Torre introduced the constrained version to their parametric blocked procedure to enhance the diagnostic power of the test by avoiding redundancy and ensuring a broader coverage of the attribute space [11]. ”

2.Can the authors list the detailed steps of the blocked NPS and GNPS methods in the main text?

Reply: 

"The proposed blocked procedures" section has been reorganized and largely rewritten to enhance clarity, with a detailed description of the steps involved in the blocked NPS and GNPS now provided in the main text. Furthermore, Fig 1 has been incorporated to visually represent the unconstrained and constrained versions of the blocked nonparametric CD-CAT procedure.

3.“how to determine the number of items that each q-vector type needed in the first step for the constrained blocked GNPS (lines 208-210)”

Reply: 

We apologize for the confusion caused by the original wording. This section has been revised to improve clarity. Since the number of q-vector types is generally larger than the block size, q-vector types are randomly selected in the constrained blocked GNPS. Each chosen q-vector type contributes one item to form the block, meaning that the number of items that each q-vector type needed is either one or zero. The revised manuscript now reads,

“In the constrained version of the blocked GNPS, since the number of distinct q-vector types M (e.g., M=7 when examining three independent attributes) typically exceeds the block size , only one item from each q-vector type is required. The items with the highest discrimination power (defined in Equation 3) within each q-vector type are identified, and the top items from this subset are selected to form the block.” (p. 14)

4.The reason of the simulation design should be explained.

Reply: 

We have rewritten the Method section to provide explanation of the simulation design in the revised manuscript on pages 16-19. 

5.During the data generation, only six attribute profiles were generated, the rests were discarded, the authors should explain this setting.

Reply: 

We regret any confusion caused by the original writing. This section "Examinee generation" has been revised to improve clarity (pp. 21-22).

The decision to focus on a subset of six attribute profiles is based on the methodology outlined by Kaplan et al. (2015), who employed a similar approach to design a more efficient simulation study. The six selected attribute profiles—ranging from no mastery to full mastery—represent key stages in attribute mastery: , ，，，，and . 

Since the attributes are assumed to be independent in our study, all attribute mastery patterns involving the same number of mastered attributes can be considered equivalent in terms of their simulation outcomes. By selecting one representative profile for each level of mastery (i.e., the number of attributes mastered), we can capture the essential dynamics of the attribute space without generating and analyzing all possible profiles. 

This more efficient simulation design allows us to significantly reduce computational time and resources while maintaining the generalizability of our findings. 

6.The authors should explain the purpose and reason of constraining the q-vector of blocked items. Did this situation (i.e., only those items with different q-vectors would be selected into the block) would be occurred in empirical study?

Reply: 

This is related to the first comment. Additionally, we have strengthened the discussion regarding the empirical study：

“The constrained version of the blocked procedure is also worth considering, as it can maintain or even enhance classification accuracy under certain conditions by constraining the q-vectors of blocked items. Since items with identical q-vectors assess the same content or skills, incorporating a variety of item types within a block is beneficial to avoid redundancy and reduce examinee boredom.”(p. 36)

7.During the simulation two (lines 375-385), which methods (package(s)) were used to calibrate item parameters? The original EM algorithm? or the Bayes Modal Estimation method (Ma & Jiang, 2021). Meanwhile, how many people were used to calibrate the items?

Ma, W., & Jiang, Z. (2021). Estimating cognitive diagnosis models in small samples: Bayes modal estimation and monotonic constraints. Applied Psychological Measurement, 45(2), 95-111.

Reply: 

We acknowledge the reference provided (Ma & Jiang, 2021). This new method enhances estimation accuracy, particularly when dealing with small sample sizes (≤100). As the sample size and test length increase, the differences between the new method and other methods become less significant. Since our focus is not on improving the calibration accuracy of parametric methods, we chose the conventional EM algorithm for simplicity. For our purpose, the use of parametric methods with small calibration samples was included to ensure a comprehensive comparison to underline the advantage of the non-parametric method over the parametric method with substantial calibration errors. 

We have clarified in the revised manuscript:

“To demonstrate the impact of small calibration samples on the performance of the parametric methods, the calibration sample size () was set to 100 [8,10].” (p. 22)

“The R package GDINA was used for item calibration in the parametric item selection strategies, specifically for the DINA and A-CDM models [32], using the marginal maximum likelihood estimation (MMLE) method with the Expectation-Maximization (EM) algorithm.” (p. 20)

We have strengthened the discussion to inform readers of the existence of new method that better calibrate items in small sample size scenarios:

“First, the nonparametric approach was introduced to address challenges associated with parametric methods, particularly in scenarios involving small calibration samples. However, recent advancements in parametric methods have yielded techniques that enhance estimation accuracy, even with small sample sizes (≤100) [36]. Future research should incorporate these developments when comparing nonparametric and parametric approaches.” (pp. 37-38)

8.During the calibration of item parameters, how to calibrate the item parameters? all 1240 items were assembled to create a single test, then calibrated using small samples (e.g., 100)? Is that realistic? Can the authors adopted the matrix sampling technique to calibrate the item parameters?

Reply: 

Thank you for your insightful comment. As this is a simulation study, it is indeed feasible to administer all 1,240 items to a calibration sample of 100 examinees during the calibration of item parameters. While this approach would be highly unrealistic in a real-world setting due to practical constraints, it is appropriate within the context of a simulation study, where the goal is to showcase the impact of calibration sample sizes on the parametric methods. In real-world applications, a matrix sampling technique would indeed be a more practical and realistic option for calibrating item parameters. However, the exploration of such techniques is beyond the scope of the current study, which is focused on the theoretical aspects of the proposed method.

9.Can the authors explain why the classification accuracy rate decreased as the block size increased?

Reply: 

Thank you for highlighting this important issue. We have accordingly expanded and strengthened the relevant section of the discussion.

It is important to note that a block size of 1 corresponds to the traditional item-level CD-CAT administration, where the test is highly adaptive, selecting the most informative item for each examinee at each stage. However, as the block size increases, the test becomes less adaptive, akin to the difference between Multi-Stage Testing (MST) and item-level CAT. In a blocked CD-CAT, multiple items are selected at once, reducing the ability to tailor each item to the examinee’s evolving ability estimate.

This reduced adaptivity is a key reason why classification accuracy tends to decrease as block size increases. Kaplan and de la Torre (2020) observed similar findings, particularly under conditions where items were of low quality and the test lengths were short or medium. 

Ultimately, the choice of block size involves a trade-off between adaptivity and other practical considerations, such as allowing students to review and change their answers. As Kaplan and de la Torre (2020) suggest, practitioners must determine the tolerable level of loss in classification accuracy when deciding on the appropriate block size for their specific context.

We have expanded our discussion on the decreased accuracy with increased block size (see p. 34-35).

10.The authors information were missing for the 18th reference in the "references" section

Reply: 

Thank you for pointing out this. We apologize for this oversight and will ensure that the correct author details are included in the revised manuscript.

Yan D, Von Davier AA, Lewis C. Computerized multistage testing: Theory and applications. New York: CRC Press; 2014.https://doi.org/10.1201/b16858

We are grateful for your careful attention to detail and the chance to refine our manuscript. We believe these revisions have greatly enhanced its clarity and coherence. Thank you again for your insightful feedback. We are confident that the manuscript is now stronger and more suitable for publication.

Response to Reviewer 2

We appreciate the time and eﬀort that you have devoted to giving us insightful suggestions and comments. We thoroughly revised our paper according to the suggestions and comments from you and the other reviewer. 

Below, we have arranged our responses below in the same sequence as your original comments/suggestions and outline changes made to the paper according to your comments/suggestions. All modiﬁcations are highlighted in the revised manuscript.

1.The simulation design used a parametric CD-CAT design, with some uncommon settings. Can the authors explain on the rationale for doing such a design, as well as the relationship to a non-parametric CD-CAT and more explanations of the simulation design? The explanations and specifications can assist the reader in understanding the distinction between non-parametric and parametric CD-CATs, conducting a non-parametric CD-CAT with proper simulation settings, and drawing the simulation results or implications. Such as, a big item pool (1240 items) is employed, which is rare in small-scale or classroom contexts. Was the sample size of 200 subjects for each studied attribute mastery pattern large enough to make an inference? Why were just the DNIA and A-CDM models used? The rationale for setting up the three levels of test lengths, etc.

a)The simulation design used a parametric CD-CAT design, with some uncommon settings. Can the authors explain on the rationale for doing such a design, as well as the relationship to a non-parametric CD-CAT and more explanations of the simulation design?

Reply: 

This important point resonates with the fourth comment from the first reviewer. We have provided an explanation of the simulation design in the revised manuscript on pages 16-19.

b)big item pool (1240 items) is employed, which is rare in small-scale or classroom contexts.

Reply: 

Thanks for bringing this out. The use of a large item pool (1240 items) in our simulation study was intentional to isolate the effects of the proposed method from those of item pool size. While we recognize that such a large item pool may be uncommon in small-scale or classroom contexts, the goal was to avoid confounding the results with the limitations that a smaller item pool could introduce. Our method is designed to work with smaller item pools as well; however, in scenarios where the it

---

## [Editor Report · Decision Letter 1]

9 Oct 2024

PONE-D-24-21730R1Introducing a Blocked Procedure in Nonparametric CD-CATPLOS ONE

Dear Dr. Li,

Thank you for submitting your manuscript to PLOS ONE. After careful consideration, we feel that it has merit but does not fully meet PLOS ONE’s publication criteria as it currently stands. Therefore, we invite you to submit a revised version of the manuscript that addresses the points raised during the review process.

After modification, I think the author has responded well to the questions of the two reviewers, and I think this paper meets the publication requirements.In addition, it is suggested that the author further supplement the relevant literature on non-parametric CDA in the introduction, such as:(1)Doi: 10.16719/j.cnki.1671-6981.20230627；(2)Doi: 10.1007/S11336-009-9125-0

We look forward to receiving your revised manuscript.

Kind regards,

Peida Zhan

Academic Editor

PLOS ONE

Journal Requirements:

Additional Editor Comments:

After modification, I think the author has responded well to the questions of the two reviewers, and I think this paper meets the publication requirements.In addition, it is suggested that the author further supplement the relevant literature on non-parametric CDA in the introduction, such as:(1)Doi: 10.16719/j.cnki.1671-6981.20230627；(2)Doi: 10.1007/S11336-009-9125-0

---

## [Author Response · Author response to Decision Letter 1]

10 Oct 2024

In this revision, we have enhanced the introduction by including relevant literature on nonparametric methods, as recommended by the editor. We would like to add the following funding information: "This research was supported by the National Key R&D Program of China (2021YFC3340801). The funders had no role in study design, data collection and analysis, decision to publish, or preparation of the manuscript." However, we could not find anywhere to enter this funding information in the submission system.

---

## [Editor Report · Decision Letter 2]

14 Oct 2024

Introducing a Blocked Procedure in Nonparametric CD-CAT

PONE-D-24-21730R2

Dear Dr. Li,

We’re pleased to inform you that your manuscript has been judged scientifically suitable for publication and will be formally accepted for publication once it meets all outstanding technical requirements.

Kind regards,

Peida Zhan

Academic Editor

PLOS ONE

Additional Editor Comments (optional):

Based on my review, the revised manuscript meets the requirements for publication and is recommended for publication. Good work.

---

## [Editor Report · Acceptance letter]

17 Oct 2024

PONE-D-24-21730R2 

PLOS ONE

Dear Dr. Li, 

I'm pleased to inform you that your manuscript has been deemed suitable for publication in PLOS ONE. Congratulations! Your manuscript is now being handed over to our production team.

Kind regards, 

on behalf of

Dr. Peida Zhan 

Academic Editor

PLOS ONE